# Natural Aporphine Alkaloids with Potential to Impact Metabolic Syndrome

**DOI:** 10.3390/molecules26206117

**Published:** 2021-10-10

**Authors:** Fei-Xuan Wang, Nan Zhu, Fan Zhou, Dong-Xiang Lin

**Affiliations:** 1Nanjing Institute of Product Quality Inspection, Nanjing 210019, China; nzhuwork@126.com (N.Z.); zhouf0623@foxmail.com (F.Z.); lindx8211@126.com (D.-X.L.); 2School of Biological Sciences & Medical Engineering, Southeast University, Nanjing 210096, China

**Keywords:** natural aporphine alkaloids, metabolic syndrome, thaliporphine, boldine, nuciferine

## Abstract

The incidence and prevalence of metabolic syndrome has steadily increased worldwide. As a major risk factor for various diseases, metabolic syndrome has come into focus in recent years. Some natural aporphine alkaloids are very promising agents in the prevention and treatment of metabolic syndrome and its components because of their wide variety of biological activities. These natural aporphine alkaloids have protective effects on the different risk factors characterizing metabolic syndrome. In this review, we highlight the activities of bioactive aporphine alkaloids: thaliporphine, boldine, nuciferine, pronuciferine, roemerine, dicentrine, magnoflorine, anonaine, apomorphine, glaucine, predicentrine, isolaureline, xylopine, methylbulbocapnine, and crebanine. We particularly focused on their impact on metabolic syndrome and its components, including insulin resistance and type 2 diabetes mellitus, endothelial dysfunction, hypertension and cardiovascular disease, hyperlipidemia and obesity, non-alcoholic fatty liver disease, hyperuricemia and kidney damage, erectile dysfunction, central nervous system-related disorder, and intestinal microbiota dysbiosis. We also discussed the potential mechanisms of actions by aporphine alkaloids in metabolic syndrome.

## 1. Introduction

The incidence and prevalence of metabolic syndrome has steadily increased worldwide [1]. Metabolic syndrome, as a clustering disorder, includes abnormal insulin and glucose metabolism, insulin resistance, hyperglycemia, hyperlipidemia, hypertension, hyperuricemia, and oxidative stress and pro-inflammatory state. Metabolic syndrome is defined as a set of risk factors for various diseases such as cardiovascular disease and type 2 diabetes, causing a significant source of morbidity and mortality [1,2]. Nowadays, in the frontier field of pharmacology, more and more studies have shown that metabolic syndrome can be managed more effectively by determining proper pharmacological approaches, such as using favorable herbs with fewer side effects [3,4].

Natural products from herbal medicines are responsible for their beneficial effects on human health, among them aporphine alkaloids are the substances of great interest [5]. Aporphine alkaloids are very promising agents in the prevention and treatment of metabolic syndrome due to a variety of pharmacological activities, particularly in anti-insulin resistance, anti-hyperlipidemia, anti-hypertension, anti-diabetes, anti-obesity, anti-oxidation, and anti-inflammation [5,6,7]. These aporphine alkaloids have preventive effects on the different risk factors characterizing metabolic syndrome, showing positive results. In this review, we highlight the activities of aporphine alkaloids: thaliporphine, boldine, nuciferine, pronuciferine, roemerine, dicentrine, magnoflorine, anonaine, apomorphine, glaucine, predicentrine, isolaureline, xylopine, methylbulbocapnine, and crebanine. We particularly focused on their impacts on metabolic syndrome and its related components, such as insulin resistance and type 2 diabetes mellitus, endothelial dysfunction, hypertension and cardiovascular disease, hyperlipidemia and obesity, non-alcoholic fatty liver disease (NAFLD), hyperuricemia and kidney damage, erectile dysfunction, central nervous system-related disorder, and intestinal microbiota dysbiosis. We also discussed their potential mechanisms of actions. This review provides the guidance for further applications of aporphine alkaloids in the prevention and treatment of metabolic syndrome and its components.

## 2. Prevention of Insulin Resistance and Type 2 Diabetes

The regulation of glucose homeostasis is an effective way to combat insulin resistance in type 2 diabetes. Aporphine alkaloids, which are natural products, have been reported to inhibit glucose uptake by isolated intestinal brush-border membrane vesicles or basolateral membrane vesicles [7]. Thaliporphine, which can be isolated from Chinese herbs such as Lauraceae, decreases glucose absorption during in situ intestinal perfusion, indicating this aporphine alkaloid has a potent effect on the inhibition of glucose uptake, which may have hypoglycemic effects [7]. Moreover, a bolus intravenous injection of thaliporphine can reduce plasma glucose level in a dose-dependent manner in both normal and streptozocin-induced diabetic rats. In diabetic rats induced by streptozotocin or nicotinamide/streptozotocin, thaliporphine is confirmed to have a potent anti-hyperglycemic effect by stimulating insulin release and increasing glucose utilization [8]. Nuciferine, a major aporphine alkaloid found in Lotus (*Nelumbo nucifera* leaves), may be a potentially important candidate in the management of insulin resistance and type 2 diabetes. The network pharmacology analysis shows that nuciferine has several biological functions, which may be associated with insulin resistance and dysregulation of lipolysis in adipocytes, which are partly confirmed [9]. Nuciferine and pronuciferine significantly increase glucose uptake in insulin resistant 3T3-L1 adipocytes, showing the amelioration of glucose metabolism. They also up-regulate glucose transporter type 4 (GLUT-4) expression, triggering the phosphorylation of activation of 5’-AMP-activated protein kinase (AMPK) in mature 3T3-L1 adipocytes [10]. Nuciferine also stimulates insulin secretion in isolated islets and insulin secreting beta cell derived line cells, possibly by closing adenosine triphosphate-sensitive potassium (KATP) channels [11]. Notably, nuciferine restores glucose tolerance impairment and insulin resistance in high-fat diets combined with streptozocin-induced diabetic mice [12]. Consistently, nuciferine decreases serum blood glucose levels in the streptozotocin-induced diabetic rat model [13].

It is known that skeletal muscle is the regulator of glucose homeostasis. *Tinospora cordifolia*, used traditionally as a blood purifier in Ayurveda, is commonly used to treat diabetes with secondary complications. *T. cordifolia* and its main constituent magnoflorine are reported to decrease the fasting blood glucose level in streptozotocin-induced diabetes in rats. They prevent a lean body, individual skeletal muscle mass, and diameter loss in myotubes. Creatine kinase (CK, as creatine phosphokinase) found in skeletal muscle, myocardium, brain, and other visceral tissues, affects mitochondrial energy metabolism. Magnoflorine decreases serum CK levels and increases myosin heavy chain-β (MyHC-β) in the muscle of animal models. Additionally, *T. cordifolia* and magnoflorine efficiently down-regulate the expression levels of ubiquitin-proteasomal E3 ligases, autophagy, and caspase-related genes in streptozotocin-induced diabetic rats [14]. Forkhead box O1 (FOXO1), mainly expressed in insulin-sensitive tissues (such as skeletal muscle), is a major target of insulin action. Analysis of molecular docking for the FOXO1 protein shows that the aporphine alkaloids anonaine, isolaureline, and xylopine isolated from *Annona muricata* leaves have an equal or smaller free binding energy compared with the control compound, with similar binding ability towards 33% amino acid residues (hydrogen bond type). This indicates that the nucleus FOXO1 protein inactivation by these compounds may have potency for the treatment of type 2 diabetes mellitus [15]. Thus, these compounds could be useful for insulin resistance and type 2 diabetic subjects. Further observations indicate that magnoflorine prevents skeletal muscle atrophy by regulating the protein kinase B (Akt)/mammalian target of rapamycin (mTOR)/FoxO signal pathway [14]. Additionally, thaliporphine stimulates the release of insulin and increases skeletal muscle glycogen synthesis in both normal and diabetic rats [8]. It also significantly attenuates an increase in plasma glucose induced by an intravenous glucose challenge test in normal rats [8]. The above observations in diabetic models suggest that the hypoglycemic effects of these aporphine alkaloids could be attributed to the stimulation of insulin release and the increase in glucose utilization in skeletal muscle.

The increased levels of oxidative stress and inflammation are suggested to accelerate the pathogenesis and progression of insulin resistance and type 2 diabetes with tissue damage. Notably, *T. cordifolia* and magnoflorine can both increase superoxide dismutase (SOD) and glutathione peroxidase (GSH-Px) activity, decrease β-glucuronidase activity, and prevent catalase activity alteration in streptozotocin-induced diabetic rats via the Akt/mTOR/FoxO signal pathway [14]. Boldine, which is found in the Chilean boldo tree, is found to decrease streptozotocin-induced elevation of mitochondria manganese-containing SOD activity in rat kidneys and pancreases. It restores the streptozotocin-induced decrease in GSH-Px activity in the liver and pancreas. Boldine attenuates streptozotocin- and iron plus ascorbate-induced methane dicarboxylic aldehyde (MDA), carbonyl formation, and thiol oxidation in the pancreas homogenates of streptozotocin-induced diabetic rats. These results show its inhibitory effect on oxidative tissue damage and antioxidant enzyme activity alteration, possibly by the decomposition of reactive oxygen species (ROS), hydrogen peroxides, hydroxyl radicals, and inhibition of nitric oxide (NO) production and peroxidation-induced product formation [16]. Boldine also reduces plasma glucose levels in both normal and streptozocin-induced diabetic rats [8]. In fact, boldine decreases glycemia and prevents against high glucose and proinflammatory cytokines induced by the decrease in gap junctional communication, the increase in oxidative stress, and cell permeability due to connexin hemichannel activity, but not the blockade of gap junction channel in mesangial cells [17]. These results indicate that boldine may attenuate streptozotocin-induced diabetes with tissue damage, possibly by interfering with oxidative stress, inflammation, and diabetes mellitus pathogeneses.

Metabolic syndrome may be a risk factor for pancreatic cancer [18,19]. Gemcitabine is a main drug against pancreatic cancer. Nuciferine can enhance the sensitivity of pancreatic cancer cells to gemcitabine in both cultured cells and the xenograft mouse model [20]. AMPK-mediated 3-hydroxy-3-methyl-glutaryl-coA reductase (HMGCR) is a rate-limiting enzyme of the mevalonate pathway for cholesterol biosynthesis. Yes-associated protein (YAP), a terminal effector of the Hippo pathway, directly interacts with sterol regulatory element binding proteins (SREBP-1c and SREBP-2) on the promoters of fatty acid synthase (FAS) and HMGCR. Nuciferine induces YAP phosphorylation at Ser127 through HMGCR down-regulation. Remarkably, wild-type YAP overexpression or the YAP Ser127 mutant could resist nuciferine and no longer sensitize pancreatic cancer cells to gemcitabine. The knockdown of AMPK attenuates nuciferine-mediated YAP phosphorylation (Ser127) and HMGCR down-regulation. Consistently, nuciferine-mediated HMGCR overexpression is observed to reduce growth inhibition in pancreatic cancer cells. These results indicate nuciferine could be an effective supplementary agent with YAP inhibition by the AMPK-mediated down-regulation of HMGCR, which may sensitize pancreatic cancer cells to gemcitabine [20].

## 3. Restoration of Endothelial Dysfunction, Hypertension, and Cardiovascular Disease

Oxidative stress in mitochondrial dysfunction plays a detrimental role in the pathogenesis of metabolic syndrome-associated endothelial dysfunction. Patients with diabetes are observed to develop endothelial dysfunction [21]. Boldine has an endothelial protective effect on hypertension and diabetes mellitus, and improves endothelial function in spontaneously hypertensive rats [22]. Consistently, boldine significantly lowers systolic blood pressure and enhances maximal relaxation to acetylcholine in spontaneously hypertensive rats. Moreover, boldine decreases aortic superoxide and peroxynitrite production, and down-regulates NADPH oxidase subunit p47(phox) protein expression in spontaneously hypertensive rat aortas [22]. These results show that boldine exerts endothelial protection in hypertension, partly through the inhibition of NADPH-mediated superoxide production. Additionally, boldine decreases blood pressure in streptozotocin-induced diabetic rats [17]. Pretreatment with boldine reduces ROS and nitrotyrosine formation, and preserves NO production in high glucose-exposed rat aortic endothelial cells. Under beta-nicotinamide adenine dinucleotide phosphate condition, it attenuates acetylcholine-driven endothelium-dependent relaxation in the cell model as well as in the aortas of streptozotocin-stimulated diabetic rats. Chronic treatment of boldine normalizes ROS overproduction with the reduction in NADPH oxidase subunit NOX2 and p47(phox) in diabetic animals. It reverses high glucose induced by the increased ROS formation in endothelial cells, and restores streptozotocin-induced endothelial dysfunction in diabetes in rats by inhibiting oxidative stress and increasing NO bioavailability [23]. Boldine also reverses high glucose or angiotensin II-induced by the impairment of relaxation in non-diabetic mouse aortas, while it reduces ROS overproduction and increases eNOS phosphorylation in db/db mouse aortas. Boldine inhibits expression of oxidative stress markers, bone morphogenic protein 4 (BMP4), nitrotyrosine, and angiotensin II type 1 receptor (AT1) in db/db mouse aortas, as well as angiotensin II-stimulated BMP4 expression [24]. It is confirmed that boldine significantly restores acetylcholine-induced endothelium-dependent relaxation in isolated thoracic aortas of spontaneously hypertensive rats, db/db mice, and streptozotocin-induced diabetic rats, respectively [25]. This endothelial protection by boldine may be correlated with an increase in vascular NO production and ROS reduction, with an angiotensin II-mediated BMP4-dependent mechanism.

Notably, an increase in lipid peroxidation and mitochondrial calcium overload are characteristics of dysfunctional endothelial cells [21]. Pretreatment with nuciferine increases endogenous antioxidant content and decreases lipid peroxidation with a significant reduction in heartbeats per minute in isoproterenol-stimulated rats [26]. In addition, nuciferine can induce relaxation in arterial segments precontracted by KCl or phenylephrine. This arterial relaxation is reduced by removal of endothelium as well as by pretreatment with endothelial NOS inhibitor L-NAME or NO-sensitive guanylyl cyclase inhibitor ODQ. In human umbilical vein endothelial cells, phosphorylation of eNOS at Ser1177 and an increase in cytosolic NO level driven by nuciferine are mediated by extracellular Ca^2+^ influx. Under endothelium-free conditions, nuciferine attenuates CaCl_2_-induced contraction in a Ca^2+^-free depolarizing medium. In the absence of extracellular calcium, nuciferine relieves vasoconstriction induced by phenylephrine and the addition of CaCl_2_. Nuciferine also suppresses Ca^2+^ influx in Ca^2+^-free K^+^-containing solution in vascular smooth muscle cells [27].

Metabolic Syndrome is a significant risk factor for myocardial infarction. The cardioprotective agent nuciferine prevents isoproterenol-induced myocardial infarction, and decreases heart weight along with lactate dehydrogenase and creatine kinase (CK-BM) cardiac marker levels in rats. It restores myocardial infarction-induced pathological implications such as tachycardia, left atrial enlargement, and anterolateral ST-elevated myocardial infarction, with the prevention of structural abnormality and inflammation in heart and liver tissues. Additionally, on in silico analysis, nuciferine shows a strong binding interaction with both β1 and β2 adrenergic receptors [26]. These results suggest that nuciferine may have a therapeutic effect on cardiovascular diseases, possibly by endothelium-dependent and -independent mechanisms.

Dicentrine, an aporphinic alkaloid found in *Stephania venosa*, is used for the treatment of cancer, diabetes mellitus, anemia, and asthma. However, importantly, dicentrine can decrease blood pressure and heart rate in both normal and spontaneously hypertensive rats under waking or anesthetic state [28]. Although dicentrine fails to diminish atherosclerotic lesion areas, it decreases mean arterial pressure significantly in spontaneously hypertensive rats and restores maximal response for phenylephrine-induced contraction in hyperlipidaemic rats [28]. The pretreatment with dicentrine and its analogs O-methylbulbocapnine and crebanineis also can restore lipopolysaccharide-induced pro-inflammatory cytokines and mediators including interleukin-6 (IL-6), tumor necrosis factor alpha (TNF-α), prostaglandin E2, and NO in RAW264.7 macrophages. Moreover, these compounds inhibit expression of inducible nitric oxide synthase (iNOS) and cyclooxygenase-2 in this cell model [29]. Dicentrine and O-methylbulbocapnine also inhibit nuclear factor kappa B (NF-κB) activation by suppressing the phosphorylation of NF-κB at Ser536, but not the nucleus translocation and inhibitor of kappaB (IκB)-α degradation in lipopolysaccharide-exposed RAW264.7 macrophages. In addition, they reduce the phosphorylation and nucleus translocation of activator protein-1 and suppress the activation of myeloid differentiation factor 88 (MyD88), Akt, as well as the upstream signaling regulator mitogen-activated protein kinases (MAPKs) signaling pathway in this cell model [29]. Therefore, dicentrine may have potential for the reduction in hypertension for cardiovascular disease.

## 4. Attenuation of Hyperlipidemia and Obesity

Hyperlipidemia is one of the diagnostic criteria for metabolic syndrome [30]. Early studies show that oral administration of dicentrine to hyperlipidaemic rats significantly reduces total cholesterol (TC), low density lipoprotein (LDL) fraction, total plasma triglyceride (TG), and very low density lipoprotein (VLDL) fraction in plasma. Moreover, dicentrine increases high density lipoprotein (HDL)-cholesterol levels, resulting in the improvement of total cholesterol to HDL-cholesterol ratio [28]. Nuciferine has the ability to alleviate dyslipidemia and decrease fat mass in vivo. Nuciferine suppresses the proliferation of 3T3-L1 preadipocytes in a dose- and time-dependent manner. It significantly reduces lipid accumulation and intracellular TG content in differentiating preadipocytes. In fully differentiated adipocytes, nuciferine down-regulates FAS, acetyl coenzyme A carboxylase (ACC), and SREBP1 mRNA levels. In fully differentiated human primary adipocytes, nuciferine also decreases FAS, ACC, and SREBP1 at mRNA levels. Notably, it reduces FAS promoter activity, which is consistent with its reduction in intracellular lipid accumulation and down-regulation of critical lipogenic enzymes in 3T3-L1 preadipocytes [31]. Furthermore, both nuciferine and pronuciferine significantly decrease lipid droplets and intracellular TC content in insulin resistant 3T3-L1 adipocytes, showing the amelioration of lipid metabolism [10]. Lipases catalyze the hydrolysis of long chain triglycerides. Recently, the lipase inhibitory kinetics of nuciferine have been determined. Its inhibitory action on lipase is reversible, showing a mixed-type inhibitor of lipase in 3T3-L1 preadipocytes. Molecular docking indicates that the interaction site between active substance and inhibitor is located in an α-helix and a β-sheet of the lipase, and this lipase active site is interfered with by inhibitor near cap structure [32]. Additionally, poly lactic-co-glycolic acid nanoparticles loaded with nuciferine are found to enhance oral-sustained and controlled drug release, with the improvement of bioavailability to alleviate lipogenesis [33].

Metabolic syndrome plays an essential role in obesity disease and the atherosclerotic process. Adipokines are secreted by adipose tissue. Adipokine FGF21 (fibroblast growth factor 21) is recently reported to improve hepatic insulin sensitivity [34]. Adipokine zinc-alpha2-glycoprotein (ZAG) is a potent lipid-mobilizing factor. Nuciferine promotes FGF21 and ZAG expression in fully differentiated adipocytes [31]. A negative correlation between oxidative stress and adiponectin (an adipokine) is observed in patients with metabolic syndrome. Boldine is able to up-regulate the expression of adiponectin regulators CCAAT/enhancer binding protein-alpha (C/EBPα) and peroxisome proliferator-activated receptor (PPAR)-gamma, and counteracts the inhibitory effect of hydrogen peroxide or TNF-α-stimulated differentiated 3T3-L1 adipocytes. Boldine also induces adiponectin at the inductive phase of adipogenesis in this cell model. More importantly, boldine interacts with the PPAR response element and could potentially modulate PPAR responsive genes [35]. This compound decreases ex vivo oxidation of LDL, and inhibits atherosclerosis in vivo in LDLR-/- mice that have been fed an atherogenic diet [36]. Therefore, nuciferine and boldine may have the potential to be beneficial in obesity disease and atherosclerosis.

Fat mass and the obesity-associated gene (FTO) may increase the risk of metabolic syndrome in subjects [37,38]. *Annona muricata* (as soursop), a common tropical plant species, has many biological activities including anti-obesity. This extract significantly down-regulates FTO via its highest binding affinity, but up-regulates the signal transducer and activator of transcription 3 (STAT-3) and reduces insulin resistance in high-fat diet-induced obesity in rats, possibly contributing to its use in the management of obesity. In silico studies show that its main alkaloids, annonaine and annonioside, have high binding affinity with FTO. Arg-96 is found to be a critical amino acid enhancing anonaine or isolaureline binding to fat mass and obesity gene, indicating that annonaine and annonioside may be probable compounds responsible for the anti-obesity effects of this herb [39].

## 5. Amelioration of NAFLD

NAFLD is considered a manifestation of metabolic syndrome. NAFLD is prevalent in liver disease associated with lipotoxicity, lipid peroxidation, oxidative stress, and inflammation. It also increases the risk of type 2 diabetes mellitus, cardiovascular disease, and chronic kidney disease [40]. In high-fat diet-fed rats, nuciferine improves hepatic steatosis, which is consistent with its reduction in body weight, blood levels of lipids and liver enzymes, and increase in SOD and GSH-Px activity, along with its decrease in MDA content in the liver, showing its anti-oxidative effect [41]. Moreover nuciferine also has anti-inflammatory effects, such as the reduction in serum and liver IL-6, interleukin-1 beta (IL-1β), and TNF-α levels [41]. Per–Arnt–Sim kinase (PASK) is a nutrient responsive protein kinase, which regulates lipid and glucose metabolism, mitochondrial respiration, and gene expression. In high-fat diet-induced NAFLD hamsters, nuciferine can protect against the increases of adipose tissue weight, dyslipidemia, and liver steatosis, and alleviate mild necroinflammation, along with reversing the serum markers of metabolic syndrome [42]. Nuciferine also down-regulates expression levels of genes related to lipogenesis, decreases free fatty acid infiltration, and up-regulates genes related to lipolysis and VLDL secretion, further demonstrating its amelioration of high-fat diet-induced dyslipidemia as well as liver steatosis and injury [42]. Nuciferine and PASK small-interfering RNA (siRNA) both inhibit TG accumulation and decrease free fatty acids (FFAs), increases total antioxidant capacity and SOD, and decreases MDA in oleic acid-exposed HepG2 cells. In addition, nuciferine decreases TNF-α, IL-6, and IL-8, and increases IL-10, as well as regulates the expression of target genes related to lipogenesis in oleic acid-exposed HepG2 cells, which is consistent with its PASK inhibition. This indicates that PASK may participate in nuciferine mediation in the prevention of NAFLD [43]. In the same basal diet *ad libitum* model of broiler chickens, a supplement of dietary nuciferine decreases body weight, average daily weight gain, and absolute and relative fat and liver weight. It also reduces plasma concentration of triiodothyronine, free triiodothyronine, thyroxine, and free thyroxine, and increases plasma concentration of glucagon in this animal. Nuciferine also decreases TG and TC levels, as well as increasing non-esterified fatty acid concentration, lipase activity, and glycogen content in the plasma and/or liver of broiler chickens [44]. This compound significantly down-regulates the gene expression level of HMGCR, SREBP2, ACC, and SPEBP-1C, but significantly up-regulates the gene expression levels of the liver X receptor α (LXR-α) and carnitine palmitoyltransferase I (CPT-I), which is consistent with its reduction in fatty degeneration. This indicates that nuciferine has the ability to reduce fat deposition by regulating lipid metabolism in broiler chickens [44].

NAFLD prevalence partially increases in line with obesity and type 2 diabetes. Nuciferine reduces hepatic levels of TC, TG, LDL, and lipid droplets in a high-fat diet combined with streptozocin-induced diabetic mice. The improvement of lipid profile and attenuation of hepatic steatosis may be associated with its activation of the liver PPARα/PPARγ coactivator-1α (PGC1α) pathway to accelerate β-oxidation [12]. Consistently, nuciferine significantly increases body weight and decreases food and water intake in streptozotocin-induced diabetic rats. Nuciferine decreases liver TC, TG, and FFA levels in this animal model, which are consistent with its down-regulation of lipogenesis and up-regulation of lipolysis and fatty acid β-oxidation [13]. Accordantly, nuciferine may be a potentially important candidate in preventing hepatic steatosis, subsequently resulting in the management of obesity and type 2 diabetes. These results may explain the anti-diabetic effect of *N. nucifera*.

NAFLD could impair bile formation and cause cholestasis. Boldine is reported to promote bile acid secretion by up-regulating the responsible transporter bile salt export pump (Bsep) and sodium/taurocholate co-transporting polypeptide (Ntcp), and preventing cholestasis by increasing glutathione secretion in rats fed a high-sucrose diet [45]. Farnesoid X receptor (FXR) is a Bsep transcriptional regulator. Recent studies show that FXR activation decreases intestinal lipid absorption and hepatic triglyceride levels to protect against NAFLD [46], and stimulating the FXR/BSEP pathway may promote the secretion of accumulated bile [47]. Boldine infusion can instantly increase the bile flow not only in healthy rats, but also in multidrug resistance-associated protein 2 (Mrp2) animals with deficiency or ethinylestradiol-induced cholestasis, without the increase in bile acid or glutathione biliary excretion. Notably, bile concentration of boldine above 10 μM is required for the induction of choleresis. In fact, Mrp2 transporter and conjugation reaction affect the biliary clearance of boldine [48]. Thus, long-term pretreatment of boldine has the potential to protect against NAFLD and then accelerate bile secretion.

## 6. Alleviation of Hyperuricemia and Kidney Damage

Hyperuricemia seriously injures human health by increasing the risk of gout. Hyperuricemia may contribute to the reduction in kidney uric acid excretion in high-fructose diet-fed rats [49]. Xanthine oxidase (XOD) can oxidize hypoxanthine to xanthine and then to uric acid. Its enzyme hyperactivity promotes the pathogenesis of hyperuricemia, therefore, XOD is a well-known target for drug development in the treatment of gout. The total alkaloids of *N. folium* are reported to markedly inhibit XOD activity. UHPLC-Q-TOF-MS and 3D docking analysis show that the natural aporphine alkaloid roemerine is a potential active ingredient [50]. Recently, nuciferine has been reported to prevent iron accumulation and lipid peroxidation, and protect against folic acid-induced acute kidney injury in mice, as well as restore RSL3-induced ferroptotic cell death in HK-2 and HEK293T cells [51]. Nuciferine also decreases serum urate levels and improves kidney function in potassium oxonate-induced hyperuricemic mice. It reverses expression alteration of renal ion transporters such as urate transporter 1 (URAT1), GLUT9, organic anion transporter 1 (OAT1), organic cation transporter 1 (OCT1), and organic cation/carnitine transporters 1/2 (OCTN1/2) in hyperuricemic mice, which is consistent with the enhancement of uric acid excretion [52]. Interestingly, nuciferine inhibits system and renal IL-1β secretion in potassium oxonate-induced hyperuricemic mice, and high-fructose diet-fed rats with metabolic syndrome. Consistently, it suppresses renal activation of Toll-like receptors, 4/myeloid differentiation factor 88/NF-kappaB (TLR4/MyD88/NF-κB) signaling, the NOD-like receptor family and pyrin domain containing 3 (NLRP3) inflammasome in these two animal models with kidney inflammation and/or proteinuria, indicating that nuciferine can prevent kidney inflammation in hyperuricemia by suppressing inflammatory signaling [49,52]. These results suggest that nuciferine may potentially help with the prevention and treatment of kidney inflammation in hyperuricemia with metabolic syndrome.

On the other hand, oxidative stress plays a detrimental role in the pathogenesis of kidney damage in renovascular hypertension. Renin-angiotensin system activation is crucial to the development and progression of hypertensive renal damage. Boldine decreases the proteinuria/creatininuria ratio, plasma thiobarbituric acid reactive substances, and slightly reduces systolic blood pressure in 2K1C hypertensive rats with the progression of kidney disease. It also decreases the alpha-smooth muscle actin (α-SMA), collagen type III (Col III), and osteopontin markers of kidney damage, and prevents high angiotensin-converting enzyme 1 (ACE-1) and transforming growth factor-beta (TGF-β) levels, resulting in prevention of kidney damage in 2K1C rats [53]. In streptozotocin-induced diabetic rats, boldine decreases renal thiobarbituric acid reactive substances and the urinary protein/creatinine ratio, and reduces the alteration of matrix proteins and markers of renal damage. Moreover in high-glucose and proinflammatory cytokines-exposed MES-13 cells, boldine prevents the increased levels of oxidative stress, decreases the number of coupled cells and cell communication via gap junction, and up-regulates connexin 43 (Cx43) in response to its protective effect [17]. These results indicate that boldine alleviates kidney damage and improves kidney function.

## 7. Recovery of Erectile Function

Penile erection is a neurovascular event driven partly by NO release from both vascular endothelial cells and neurons. NO function acts as a vasodilator causing penile engorgement and erection. Erectile dysfunction, a common disease, is characterized by endothelial dysfunction and is also closely related to cardiovascular morbidity and mortality. Its etiology is often multifactorial biochemical perturbations. Uric acid causes endothelial dysfunction via the reduction in NO production, leading to microvascular changes. Clinical evidence shows that hyperuricemia can be considered a risk predictor for erectile dysfunction [54]. Resistin, an adipokine, is known as a potential mediator of obesity and associated insulin resistance. Its high level disrupts NO-mediated relaxation. Animal studies show that metabolic syndrome induces a deterioration in erectile function with the over expression of resistin and suppression of endothelial nitric oxide synthase (eNOS) activity in rat penile tissue. Diabetes mellitus has a negative impact on blood flow in cavernosal tissue through activating vasoconstrictor mediators. Endothelin receptors and their relationship with eNOS may impair erectile response in diabetic rats [55]. Therefore, metabolic syndrome or its components are associated with erectile dysfunction [56].

Natural aporphine alkaloids could be new options for treating erectile dysfunction in patients with diabetes mellitus. Apomorphine can stimulate the central neurogenic pathway. Sublingual apomorphine is considered as a suitable alternative in diabetic patients with erectile dysfunction [57]. In addition to its anti-diabetic effect, boldine protects endothelial function and enhances intracavernous pressure/mean arterial pressure value during cavernous nerve stimulation, insulin receptor-β, and p(S1177) eNOS expression in the penile tissue of the animal metabolic syndrome model with insulin resistance, showing its improvement of erectile function, independent of resistin expression [58]. Diallyl thiosulfinate (allicen) displays anti-oxidant, anti-inflammatory, lipid-modifying, anti-obesity, antihypertensive, and anti-diabetic effects on metabolic syndrome risk factors (hyperlipidemia, high blood glucose, obesity, and hypertension) [59,60,61]. Sterol saponin diosgenin is reported to exert beneficial effects on glucose tolerance, insulin action, inflammation, blood lipid levels, and cardiovascular health. Diosgenin attenuates the effects of high-fat diet- or/and high-sugar diet-induced metabolic syndrome in rodents [62,63]. According to the International Index of Erectile Function-5 score, a single blind study shows that diallyl thiosulfinate with nuciferine and diosgenin as oral tablets are able to improve the control of ejaculation in patients suffering from premature ejaculation, primary or secondary to erectile dysfunction [64]. These results suggest that the therapy of these natural products is able to treat premature ejaculation associated with metabolic syndrome.

## 8. Improvement of Brain Function

Alzheimer’s disease (AD) is the most common cause of senile dementia worldwide, characterized by both cognitive and behavioral deficit. Evidence has shown that metabolic syndrome drives blood-brain barrier impairment with neuroinflammation development paralleled by the accumulation of toxic amyloid, causing cognition impairment in different cognitive domains [65,66]. Aβ oligomers (AβO) are responsible for several pathological mechanisms during the development of AD, including abnormality of cellular homeostasis and synaptic function, inevitably leading to cell death. Boldine has the capacity to block dysfunctional processes caused by AβO. It can interact with Aβ in silico affecting its aggregation and protecting AβO-induced synaptic failure in primary hippocampal neurons. Boldine also normalizes changes in intracellular Ca^2+^ levels associated to mitochondria or endoplasmic reticulum in the HT22 hippocampal-derived cell line exposed by AβO. In addition, boldine attenuates the AβO-induced decrease in mitochondrial membrane potential and mitochondrial respiration as well as the increase in mitochondrial ROS in HT22 hippocampal cells. Therefore, boldine exhibits neuroprotection in an AD model by both direct interaction with Aβ and prevention of mitochondrial dysfunction [67]. Moreover, boldine can improve the learning and memory of Swiss albino male young and aged mice, possibly by suppressing brain acetylcholinesterase activity and oxidative stress [68]. Its neuroprotection may be mediated by the attenuation of neuroinflammation and memory deficit induced by permanent middle cerebral artery occlusion in mice [69].

Recently, AD has been considered as ‘diabetes of the brain’ or ‘type 3 diabetes’. Clinical trials of anti-Aβ therapy have not proved to be successful. Insulin resistance is detected in neurons of AD. Insulin-degrading enzyme (IDE) is linked to insulin signaling and degrades Aβ. Apomorphine can promote intraneuronal Aβ degradation and improve memory function in a triple-transgenic mouse model of AD (3xTg-AD) [70]. One monthly subcutaneous injection of Apokyn^®^ to 3xTg-AD mice at 6 or 12 months of age improves memory function significantly. This injection effectively increases brain IDE in 3xTg-AD mice, and down-regulates brain protein levels of serine-phosphorylated insulin receptor substrate-1 (IRS-1), pS616 and pS636/639 in 13-month-old 3xTg-AD mice, respectively [70,71]. Therefore, brain insulin resistance may be an important therapeutic target in AD. Recovery of neuronal insulin signaling impairment by aporphine alkaloids may be a promising therapeutic strategy for AD dementia. However, additional studies are required to evaluate the effect of aporphine alkaloids on cognitive and behavioral deficit with insulin resistance and metabolic syndrome.

Nuciferine is reported to have neuroprotective and therapeutic effects in cerebrovascular diseases. Nuciferine significantly improves the neurological deficit score and ameliorates cerebral edema and infarction in middle cerebral artery occlusion-induced strokes of rats, showing its anti-stroke effect [72]. Furthermore, multivariate data analysis shows that nuciferine treatment may mediate approximately 19 metabolites and 7 metabolic pathways involved in neuroprotection, anti-apoptosis and anti-inflammation to restore metabolic disturbances [72]. These observations indicate that nuciferine has a positive therapeutic action on cerebral ischemia, possibly by regulating its metabolism.

Roemerine, a quaternary ammonium isoquinoline alkaloid isolated from *P. lacerum* and *P. syriacuma*, has antidepressant-like effects. This constituent exhibits neuronal activity through increasing brain-derived neurotrophic factor protein expression and affecting the serotonergic and glutamatergic systems in the SH-SY5Y cell line [73]. Berberine, a quaternary ammonium isoquinoline alkaloid, has potentially beneficial effects on dementia type 2 diabetes and dyslipidemia related to metabolic syndrome [74,75]. Growth rate and checkerboard assay show the synergy of roemerine and berberine, the former can prevent berberine efflux by inhibiting Bmr belonging to the major facilitator and ATP-binding cassette superfamilies [75]. Conjugation of roemerine to the substrate of efflux pump may help to potentiate the activity of drug substrates. However, large-scale clinical trials may be warranted for the promising synergy of these compounds on brain dysfunction of metabolic syndrome.

Multiple sclerosis is a chronic, inflammatory, demyelinating disease of the central nervous system (CNS). High frequency of metabolic syndrome and its components are observed in patients with multiple sclerosis [76]. As the most abundant gap junction protein in CNS, astrocytic Cx43 maintains astrocyte network homeostasis, affects oligodendroglial development, and participates in CNS pathology. Notably, acceleration of remyelination process and modulation of local inflammation are observed in astrocyte-specific Cx43 conditional knockout mice. Boldine blocks the Cx43 hemichannel activity in astrocytes without affecting gap junctional communication, and suppresses local inflammation while enhancing remyelination [77]. Thus, the inhibition of the Cx43 hemichannel function by boldine may be a potential therapeutic approach for demyelinating diseases in metabolic syndrome.

## 9. Potent Restoration of Intestinal Microbiota-Mediated Metabolic Syndrome

Gut microbiota dysbiosis as well as gut microbiota-liver axis dysfunction are the contributing factors to the pathogenesis of metabolic syndrome [78,79,80]. Intestinal microbiota may be a novel drug target of metabolic syndrome, especially for those with poor oral bioavailability. Nuciferine in lotus extracts is absorbed by the intestinal tract, showing a low permeability under a multi-component environment [81,82]. Nuciferine supplementation can prevent weight gain, reduce fat accumulation, and ameliorate lipid metabolic disorder in high-fat diet-fed rats [83]. Antibiotics substantially abolish the beneficial anti-hyperlipidemic and anti-liver steatosis effects of nuciferine. 16S rRNA gene sequencing of fecal microbiota further shows that nuciferine also modulates gut microbiota dysbiosis and changes the diversity and composition of gut microbiota towards a healthy level in high-fat diet-fed rats [83]. The data from the composition of gut microbiota show that nuciferine significantly shifts microbial structure, which includes the reduction in the abundance of Butyricimonas and the enhancement of the abundance of Akkermansia, an anti-obesity bacterium in high-fat diet-fed animals [84,85]. These results indicate that gut microbiota may play an essential role in the anti-hyperlipidemic and liver steatosis-alleviating effects of nuciferine.

Abundant short chain fatty acids (SCFAs) are produced by gut microbiota, possibly suppressing the development of metabolic syndrome. The increasing levels of microbiome-modulated metabolites such as lipopolysaccharides (endotoxin, E. coli lipopolysaccharide) may deregulate the gut endothelial barrier function, facilitating metabolic disorders such as NAFLD and hyperuricemia [86,87]. Nuciferine is reported to decrease the ratio of phyla Firmicutes/Bacteroidetes, the relative abundance of lipopolysaccharide-producing genus Desulfovibrio and bacteria in lipid metabolism, whereas it increases the relative abundance of SCFA-producing bacteria in high-fat diet-fed rats with NAFLD [83]. Predicted functional analysis of the microbial community has found that nuciferine modifies genes related to lipopolysaccharide biosynthesis and lipid metabolism. In addition, nuciferine effectively improves high-fat diet-induced serum disorder of endogenous metabolism, especially lipid metabolism. Notably, nuciferine promotes SCFA production and enhances intestinal integrity, resulting in the reduction in blood endotoxemia which suppresses inflammation in high-fat diet-fed rats, which are consistent with its modulation of gut microbiota [83]. Furthermore, nuciferine treatment for 12 weeks can improve dysbacteriosis (increases the relative abundance of Alloprevotella, Turicibacter, and Lactobacillus, lowers the relative abundance of Helicobacter), as well as decrease IL-6, IL-1β, and TNF-α levels in adipose tissue or serum and up-regulate the tight junction-related genes occludin and zonula occludens 1 (ZO-1) in colon tissue in high-fat diet-fed mice [88]. Nuciferine also reduces weight gain, fat accumulation, and intestinal permeability in high-fat diet-fed C57BL/6J mice, with the improvement of autophagy. Subsequently, in lipopolysaccharide-exposed Caco-2 and HT-29 cells, this compound promotes the formation of autophagosomes and autophagolysosomes and alleviates intestinal permeability reduction. Importantly, it protects against lipopolysaccharide-induced paracellular permeability impairment after the transfection of the autophagy-related gene (*Atg*) 5 siRNA, indicating that nuciferine improves autophagy with intestinal permeability [84]. On the other hand, metabolomic analysis further characterizes the effects of nuciferine for NAFLD at a metabolic level. Nuciferine can improve metabolism disorders (such as dysfunction of glycerophospholipid, linoleic acid, α-linolenic acid, arginine, and proline metabolism) in the NAFLD rat model, with the regulation of liver gene expression of key enzymes related to the metabolism pathways mentioned above [41,42,43].

Notably, in potassium oxanate-induced hyperuricemia in rats, the non-targeted metabolomics by ^1^H NMR and liquid chromatography-mass spectrometry show that a total of 21 metabolites are authenticated in plasma and urine. These metabolites are mainly correlated to glycine, serine, and threonine metabolism, synthesis and degradation of ketone bodies, butanoate metabolism, pyruvate metabolism, citrate cycle, glycolysis/gluconeogenesis, glyoxylate and dicarboxylate metabolism, and glycerophospholipid metabolism [87]. 16S rRNA analysis indicates that diversified intestinal microorganisms may be closely related to changes in differential metabolites, especially bacteria from *Firmicutes* and *Bacteroidetes*. Among them, uric acid as well as indoxyl sulfate and *N*-acetylglutamate in urine may be potential biomarkers for the prevention of hyperuricemia. Gut microbiota changes are also related to these metabolites. More importantly, nuciferine may restrain the pathological process of hyperuricemia in rats by regulating metabolic pathway perturbation and gut microbiota composition [87]. Therefore, the anti-NAFLD or anti-hyperuricemia effects of nuciferine may be related to modulation in the composition and potential function of gut microbiota, improvement of intestinal barrier integrity, and prevention of chronic low-grade inflammation. These results may provide support for the application of nuciferine in the prevention and treatment of these metabolic disorders.

Thaliporphine significantly reduces serum superoxide anion and TNF-α levels and increases the late-phase decrease in blood glucose in lipopolysaccharide-induced endotoxaemia in rats. It attenuates endotoxaemia-induced multiple organ injury in the liver, kidney, and heart, with the reduction in serum aspartate aminotransferase, alanine aminotransferase, creatinine, lactate dehydrogenase, and CK-MB levels [89]. Thaliporphine can recover the impairment of left ventricular systolic function after lipopolysaccharide injection, which is consistent with the reduction in LDH concentration, as well as TNF-α and caspase3 dependent cell apoptosis. Further study shows that thaliporphine protects against myocardial dysfunction both preload-dependently and –independently in lipopolysaccharide-induced endotoxemic rabbits, possibly by up-regulating the PI3K/Akt/mTOR pathway and down-regulating the p38 MAPK/NF-κB pathway [90]. These observations indicate that thaliporphine may be a novel agent for attenuating endotoxin-induced metabolic disorders with organ injury.

## 10. Potential Effect on Other Disorders

Mounting evidence shows that metabolic syndrome and insulin resistance are associated with worsened outcomes of chronic lung disease [91], as well as microbiome dysbiosis notably in the airways of patients with asthma [92]. Boldine is predicted to be potential drug for asthma treatment [93]. Nuciferine can induce relaxation in contracted tracheal rings. Nuciferine relaxes high K^+^-contracted mouse tracheal rings and inhibits extracellular Ca^2+^ influx and voltage-dependent L-type Ca^2+^ channel currents. It restores acetylcholine-induced contraction in mouse tracheal rings with intracellular Ca^2+^ influx and whole-cell currents of non-selective cation channels. Thus, nuciferine may have a therapeutic effect on respiratory disorders associated with aberrant contractions of airway smooth muscles and/or bronchospasm [94].

Gastrointestinal disorder is becoming prevalent in a large part of the world’s population, and has a link to metabolic syndrome [79]. Peumusboldus Molina is widely used in traditional medicine for the treatment of digestive disorders. Its main alkaloid boldine combats symptoms of gastrointestinal disorders [95,96]. The function impairment of the 5-hydroxytrptamine 3 (5-HT3) receptor is involved in the pathogenesis of gastrointestinal disorders. Boldine inhibits 5-HT-induced activation of 5-HT3 receptors in human colon tissue [95]. Boldine is also able to protect against ethanol/HCl or indomethacin-induced gastric mucosa damage, with the reduction in oxidative stress and inflammatory mediators in mice, possibly being dependent on non-protein sulfhydryl groups and prostanoids [96]. Thus, boldine with the gastroprotective effect may be used to treat gastrointestinal disorders associated with metabolic syndrome.

Clinical study has shown that long-term metabolic syndrome has a significant relationship to periodontitis in adults under 45 years old and to periodontal pockets and alveolar bone levels [97,98]. Boldine inhibits alveolar bone resorption by modulating T-helper type-17 (Th17) and T-regulatory (Treg) lymphocyte activity imbalance in ligature-induced periodontitis in mice. Consistently, boldine produces a reduction in the osteoclast number as well as pro-osteolytic factor termed receptor activator of nuclear factor κB ligand (RANKL) and the ratio of its competitive antagonist osteoprotegerin in periodontal lesions. It reduces Th17-lymphocyte detection and response and increases Treg-lymphocyte detection and response in periodontitis-affected tissues [99].

Osteoporosis associated with a loss of bone mineral density is an enormous health problem. There is the relationship between metabolic syndrome and bone mineral density [100,101,102]. The data from the Micro-CT and histomorphometry assay show that boldine conducts a protective effect for estrogen deficiency-induced bone loss in mice through inhibiting bone resorption, without affecting bone formation in vivo. Boldine also inhibits RANKL-induced osteoclast formation via regulating the AKT signaling pathway [103]. In addition, rheumatoid arthritis, a chronic autoimmune inflammatory disease, is associated with multiple metabolic alterations [104]. Boldine reduces ankle swelling, alleviates pathological damage, and prevents bone destruction in collagen-induced arthritis in rats. Consistently, boldine decreases serum tartrate-resistant acid phosphatase (TRACP) 5b levels and osteoclast number in the ankle region from collagen-induced arthritis in rats. This compound can up-regulate expression of osteoprotegerin, and down-regulate expression of RANK and its ligand RANKL in this animal model. Thus, boldine inhibits osteoclastogenesis and alleviates bone destruction, possibly by inhibiting the RANK/RANKL signaling pathway [105]. Nuciferine also alleviates osteoclastogenesis and bone resorption. Platelet derived growth factor-BB (PDGF-BB) could help treat osteoporosis. Type H vessels are reported to couple angiogenesis and osteogenesis during osteoclastogenesis. Nuciferine restrains expression levels of osteoclast-specific genes and proteins, promotes PDGF-BB production and potentiates angiogenic activity by the inhibition of MAPK and the NF-κB signaling pathway in vitro. In the femur of ovariectomized mice, nuciferine also increases PDGF-BB concentration, decreases multinucleated osteoclast formation, and promotes type H vessel formation, resulting in its inhibition of osteoclastogenesis and prevention of bone loss [106]. Therefore, aporphine alkaloids, boldine and nuciferine, may be excellent agents in controlling the risk of metabolic syndrome to treat periodontitis, osteoporosis, or rheumatoid arthritis, but further research is needed to prove these actions.

## 11. Conclusions

Metabolic syndrome is considered as a multisystem disease that affects functions of tissues and organs. It increases the risk of cardiovascular diseases, type 2 diabetes mellitus, and chronic kidney disease, hence, metabolic syndrome has become a major health hazard in modern society. Therefore, the discovery of novel therapeutic drugs for the prevention and treatment of metabolic syndrome are urgently required. This review provides evidence that the anti-metabolic syndrome impacts of natural aporphine alkaloids from healthy herbs have been greatly extended in recent years (Table 1). Insulin resistance is the major underlying mechanism responsible for metabolic syndrome. Notably, almost all aporphine alkaloids in this review can prevent insulin resistance or/and regulate glucose homeostasis, of which boldine and nuciferine have potential pharmacological effects on metabolic syndrome components (such as endothelial dysfunction, hypertension and cardiovascular disease, hyperlipidemia and obesity, non-alcoholic fatty liver disease, hyperuricemia and kidney damage, erectile dysfunction, central nervous system-related disorder, and intestinal microbiota dysbiosis). Thus, the aporphine core structure with methoxyl or/and hydroxyl substitutions at its C1, C2, C9, or/and C10 positions may be important for these activities (Figure 1). However, the structure-activity profiling of natural aporphine alkaloids against metabolic syndrome and its components should be addressed in the future. The therapeutic approaches of aporphine alkaloids can modulate oxidative stress and inflammation in the context of metabolic syndrome. Additionally, a key question that remains to be answered is whether these aporphine alkaloids act directly or indirectly on metabolic syndrome. For example, the gut-microbiota has been identified as a key modifier, increasing its recognition as another indirect pathway for oral administration of natural aporphine alkaloids to act on CNS, heart, liver, and kidney. Therefore, it is important to investigate these possibilities, focusing on aporphine alkaloids, to help identify relevant mechanisms. Further pharmacology and toxicology research of these aporphine alkaloids, as well as the availability of therapeutic options for metabolic syndrome, will hopefully curb the rising trend of metabolic syndrome-related diseases.

## Figures and Tables

**Figure 1 molecules-26-06117-f001:**
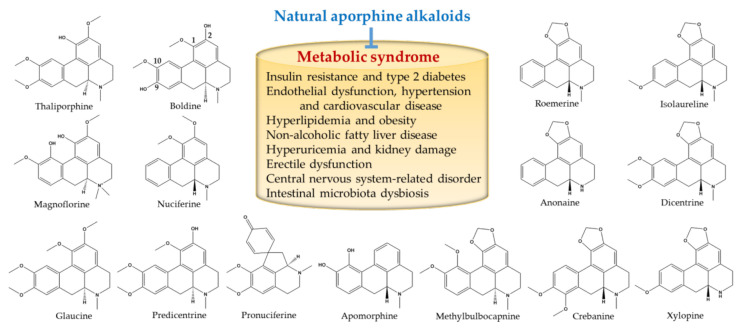
The structure of natural aporphine alkaloids with potential to impact metabolic syndrome.

**Table 1 molecules-26-06117-t001:** The potential effect of natural aporphine alkaloids on metabolic syndrome-related disorders.

Compounds	Model	Function and Possible Molecular Mechanisms	Ref.
Prevention of insulin resistance and type 2 diabetes			
Thaliporphine	In situ rat intestinal perfusion.	Decrease in intestinal glucose absorption.	[7]
Thaliporphine	Diabetic rats induced by streptozocinor nicotinamide/streptozotocin.	Reduction in plasma glucose level, stimulation of insulin release, increase in skeletal muscle glycogen synthesis.	[8]
Nuciferine	Isolated islets and insulin secreting beta cell derived line cells.	Stimulation of insulin secretion, closing KATP channel.	[11]
Nuciferine	High-fat diet combined with streptozocin-induced diabetic mice.	Restoration of glucose tolerance impairment and insulin resistance.	[12]
Nuciferine	Streptozotocin-induced diabetic rats.	Decrease in serum blood glucose level.	[13]
Magnoflorine	Streptozotocin-induced diabetic rats.	Decrease in fasting blood glucose level, prevention of skeletal muscle atrophy, increase in SOD and GSH-Px activity, decrease in β-glucuronidase activity, prevention of catalase activity alteration, regulation of Akt/mTOR/FoxO signaling, down-regulation of ubiquitin-proteasomal E3-ligases, and autophagy.	[14]
Boldine	Streptozotocin-induced diabetic rats.	Restoration of GSH-Px activity change, inhibition of oxidative tissue damage and antioxidant enzyme activity alteration in liver and pancreas, attenuation of MDA, carbonyl formation and thiol oxidation in pancreas homogenates.	[16]
Boldine	Streptozocin-induced diabetic rats	Reduction in plasma glucose level	[8]
Nuciferine Pronuciferine	Insulin resistant 3T3-L1 adipocytes.	Increase in glucose uptake, up-regulation of GLUT-4, triggering the phosphorylation of activation of AMPK.	[10]
Anonaine Isolaureline Xylopine	Molecular docking.	Nucleus FOXO1 protein inactivation. Binding ability towards 33% amino acid residues (hydrogen bond type).	[15]
Restoration of endothelial dysfunction, hypertension, cardiovascular disease			
Boldine	Spontaneously hypertensive rats.	Endothelial protection of hypertension and diabetes mellitus, improvement of endothelial function, decrease in aortic superoxide and peroxynitrite production, down-regulation of p47(phox).	[22]
Boldine	Streptozotocin-induced diabetic rats.	Reduction in blood pressure.	[17]
Boldine	Streptozotocin-induced diabetic rats.	Attenuation of endothelial dysfunction andROS overproduction, inhibition of oxidative stress, increase in NO bioavailability, reduction in NOX2 and p47(phox).	[23]
Boldine	Non-diabetic mice.	Restoration of high glucose or angiotensin II-induced impairment of relaxation in aortas.	[24]
Boldine	db/db mice.	Reduction in ROS overproduction and increase in eNOS phosphorylation in aortas, suppression of BMP4, nitrotyrosine and AT1 in aortas.	[24]
Boldine	Spontaneously hypertensive rats, db/db mice and streptozotocin-induced diabetic rats.	Restoration of acetylcholine-induced endothelium-dependent relaxation in isolated thoracic aortas.	[25]
Nuciferine	Isoproterenol-stimulated rats.	Increase in endogenous antioxidant content and decrease in lipid peroxidation	[26]
Nuciferine	Rats or vascular smooth muscle cells.	Vasorelaxant effect in rings of main mesenteric arteriesphosphorylation of eNOS at Ser1177 and increase in cytosolic NO level, suppression of Ca^2+^ influx	[27]
Nuciferine	Isoproterenol-induced rats.	Prevention of myocardial infarction, decrease in heart weight, cardiac markers lactate dehydrogenase, and CK-BM levels.	[26]
Dicentrine	Lipopolysaccharide-exposed RAW264.7 Macrophages.	Anti-inflammation.	[29]
Attenuation of hyperlipidemia and obesity			
Nuciferine	3T3-L1 preadipocytes.	Inhibition of proliferation and differentiation of 3T3-L1 preadipocytes.	[31]
Nuciferine	3T3-L1 preadipocytes molecular docking.	A mixed-type inhibitor of lipase located in an α-helix and a β-sheet.	[32]
Nuciferine	Fully differentiated adipocytes	Promotion of FGF21 and ZAG expression.	[31]
Nuciferine Pronuciferine	Insulin resistant 3T3-L1 adipocytes.	Suppression of proliferation of 3T3-L1 preadipocytes, amelioration of lipid metabolism.	[10]
Boldine	3T3-L1 cells.	Modulation of adiponectin expression and its regulators.	[35]
Boldine	In vitro and atherosclerosis in vivo in LDLR(-/-) mice.	Decrease in ex-vivo oxidation of LDL, inhibition of atherosclerosis.	[36]
Anonaine Isolaureline	High-fat diet-induced obese rats.	Down-regulation of FTO.	[39]
Amelioration of NAFLD			
Nuciferine	High-fat diet-fed rats.	Alleviation of hepatic steatosis.	[41]
Nuciferine	High-fat diet-induced dyslipidemia hamsters	Prevention of hepatic steatosis, adipose tissue weight, dyslipidemia, alleviation of mild necroinflammation, restoration of serum markers of metabolic syndrome.	[42]
Nuciferine	Oleic acid-exposed HepG2 cells.	Inhibition of TG accumulation, decrease in FFAs, increase in total antioxidant capacity and SOD, reduction in MDA, TNF-α, IL-6, and IL-8, increase in IL-10.	[43]
Nuciferine	Broiler chickens.	Reduction in fat deposition, plasma concentration of triiodothyronine, free triiodothyronine, thyroxine, and free thyroxine, and increase in plasma glucagon concentration.	[44]
Nuciferine	Streptozocin-induced diabetic mice combined with high-fat diet.	Reduction in hepatic TC, TG, and LDL levels, lipid droplets.	[12]
Nuciferine	Streptozotocin-induced diabetic rats.	Decrease in liver TC, TG, and FFAs levels.	[13]
Boldine	High-sucrose diet-induced hereditary hypertriglyceridemic rats.	Increase in biliary glutathione secretion and attenuation of cholestasis associated with non-alcoholic fatty liver disease.	[45]
Alleviation of hyperuricemia and kidney damage			
Roemerine	UHPLC-Q-TOF-MS and 3D docking analysis.	Inhibition of XOD activity.	[50]
Nuciferine	Mic with folic acid-induced acute kidney injury.	Mitigation of pathological alterations, amelioration of inflammatory cell infiltration and kidney dysfunction.	[51]
Nuciferine	HK-2 and HEK293T cells.	Inhibition of RSL3-induced ferroptosis	[51]
Nuciferine	Potassium oxonate-induced hyperuricemic mice.	Decrease in serum urate levels, improvement of kidney function, attenuation of expression alteration of renal ion transporters.	[52]
Boldine	2K1C hypertensive rats.	Reduction in ACE-1 and TGF-β levels, alleviation of kidney damage.	[53]
Boldine	Streptozotocin-induced diabetic rats.	Prevention of the increased levels of glycemia, blood pressure, renal thiobarbituric acid reactive substances, and urinary protein/creatinine ratio.	[17]
Boldine	High glucose and proinflammatory cytokines- induced MES-13 cells.	Reduction in oxidative stress, improvement of gap junctional communication, and cell permeability.	[17]
Recovery of erectile function			
Boldine	Metabolic syndrome rats.	Enhancement of intracavernous pressure/mean arterial pressure value, improvement of erectile function.	[58]
Nuciferine with diallyl thiosulfinate and diosgenin	Patients with premature ejaculation.	Improvement of the control of ejaculation, and erectile dysfunction.	[64]
Improvement of brain function			
Boldine	Primary hippocampal neurons and HT22 hippocampal-derived cell line treated with AβO.	Interaction with Aβ, prevention of oxidative stress and mitochondrial dysfunction, neuroprotection.	[67]
Boldine	Swiss albino male young and aged mice.	Improvement of learning and memory.	[68]
Boldine	Permanent middle cerebral artery occlusion mice.	Decrease in the infarct area, improvement of neurological scores, increase in cell viability.	[69]
Nuciferine	Rats with middle cerebral artery occlusion.	Improvement of neurological deficit score and amelioration of cerebral edema and infarction.	[72]
Potent restoration of intestinal microbiota-mediated metabolic syndrome			
Nuciferine	High-fat diet-fed rats with NAFLD.	Prevention of weight gain, reduction in fat accumulation and amelioration of lipid metabolic disorder, change of the diversity and composition of gut microbiota, promotion of SCFA production, and enhancement of intestinal integrity.	[83]
Nuciferine	High-fat diet obese mice.	Alleviation of dysbacteriosis, decrease in IL-6, IL-1β, and TNF-α levels in adipose tissue or serum.	[88]
Nuciferine	Lipopolysaccharide-exposed Caco-2 and HT-29 cells.	Promotion of the formation of autophagosomes and autophagolysosomes, alleviation of intestinal permeability reduction, improvement of autophagy with intestinal permeability.	[84]
Nuciferine	Rats with potassium oxanate-induced hyperuricemia.	Restrain of the pathological process of hyperuricemia.	[87]
Thaliporphine	Rats with lipopolysaccharide-induced endotoxaemia.	Reduction in serum superoxide anion and TNF-α levels, increase in late-phase decrease in blood glucose, attenuation ofendotoxaemia-induced multiple organ injury in the liver, kidney and heart.	[89]
Thaliporphine	Lipopolysaccharide-induced endotoxemic rabbits.	Recover of the impairment of left ventricular systolic function.	[90]
Possible effect on other disorders			
Boldine	Asthmatic patients.	Potential drug for asthma treatment.	[93]
Boldine	Patients with functional gastrointestinal disorders.	Inhibition of 5-HT-induced activation of 5-HT3 receptor, alleviation of functional gastrointestinal disorders.	[95]
Boldine	Ethanol/HCl or indomethacin-induced mice.	Protection of against gastric mucosa damage, reduction in oxidative stress, and inflammatory mediators.	[96]
Boldine	Mice with ligature-induced periodontitis.	Inhibition of the alveolar bone resorption and modulation of the Th17/Treg imbalance.	[99]
Boldine	Estrogen deficiency-induced osteoporosis mice. RANKL-induced osteoclast formation.	Prevention of osteoporosis by inhibiting bone resorption. Regulation of AKT signaling.	[103]
Boldine	Rats with collagen-induced arthritis.	Reduction in ankle swelling, alleviation of pathological damage, prevention of bone destruction, inhibition of RANK/RANKL signaling.	[105]
Nuciferine	Ovariectomized mice.	Preservation of Trap+ preosteoclasts, decrease in multinucleated osteoclast formation, promotion of type H vessel formation, suppression of MAPK and NF-κB signaling, inhibition of osteoclastogenesis and prevention of bone loss.	[106]
Nuciferine	High K^+^-contracted mice.	Induction of relaxation in contracted tracheal rings.	[94]
Nuciferine	Acetylcholine-stimulated mice.	Inhibition of tracheal rings.	[94]

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
