# Peer review of "Natural Aporphine Alkaloids with Potential to Impact Metabolic Syndrome"

_molecules, 2021, doi:10.3390/molecules26206117_

Round 1

Reviewer 1 Report

In “Natural Aporphine Alkaloids with Potential to Impact Metabolic Syndrome”, Fei-Xuan Wang et al highlighted the activities of several bioactive aporphine, authors had a particular focus on their impact on metabolic syndrome and its components, including insulin resistance and type 2 diabetes mellitus, endothelial dysfunction, hypertension and cardiovascular disease, hyperlipidemia and obesity, non-alcoholic fatty liver disease, hyperuricemia and kidney damage, erectile dysfunction, central nervous system-related disorder, intestinal microbiota dysbiosis, as well as chronic lung disease, gastrointestinal disorder, periodontitis, osteoporosis, and/or rheumatoid arthritis. However, there are several points:

  1. The title refers natural aporphine alkaloids with potential to impact metabolic syndrome; the association with chronic lung disease, gastrointestinal disorder, periodontitis, osteoporosis, and rheumatoid arthritis, is not clear.
  2. It is recommendable include several figures or schemes about the topics.
  3. Is recommendable focus only in metabolic syndrome, and discuss meticulously about the molecular mechanisms that aporphine alkaloids regulate.
  4. There is a structural association about the protein-targets that are regulated? An analysis is desirable.
  5. It is desirable including a scheme with the structures of analyzed aporphine alkaloids.

Author Response

Molecules Editors

Sep. 24, 2021

Re: molecules-1366483

Dear Editors,

We thank you very much for giving us an opportunity to revise our manuscript titled by ‘Natural Aporphine Alkaloids with Potential to Impact Metabolic Syndrome’ (No: molecules-1366483). We appreciate editor and reviewers for the positive and constructive comments and suggestions on our manuscript. We have carefully revised the manuscript according to the comments and suggestions. Enclosed please see point-by-point response to the reviewers' comments and suggestions as flowing.

In response to Comments and Suggestions for Authors
-Reviewer 1

In “Natural Aporphine Alkaloids with Potential to Impact Metabolic Syndrome”, Fei-Xuan Wang et al highlighted the activities of several bioactive aporphine, authors had a particular focus on their impact on metabolic syndrome and its components, including insulin resistance and type 2 diabetes mellitus, endothelial dysfunction, hypertension and cardiovascular disease, hyperlipidemia and obesity, non-alcoholic fatty liver disease, hyperuricemia and kidney damage, erectile dysfunction, central nervous system-related disorder, intestinal microbiota dysbiosis, as well as chronic lung disease, gastrointestinal disorder, periodontitis, osteoporosis, and/or rheumatoid arthritis.

However, there are several points:

  1. The title refers natural aporphine alkaloids with potential to impact metabolic syndrome; the association with chronic lung disease, gastrointestinal disorder, periodontitis, osteoporosis, and rheumatoid arthritis, is not clear.

Reply: Thank you for the comments. As suggested, we deleted ‘the association with chronic lung disease, gastrointestinal disorder, periodontitis, osteoporosis, and rheumatoid arthritis’ in the Abstract and Induction. In fact, metabolic syndrome is defined as a set of risk factors for various diseases including chronic lung disease, gastrointestinal disorder, periodontitis, osteoporosis, and rheumatoid arthritis. In the text of the revised manuscript, we revised the ‘10. Possible effect on other disorders’, shortened or changed the sentence as well as discussed the possible effect of aporphine alkaloids on other disorders associated with metabolic syndrome.

  1. It is recommendable include several figures or schemes about the topics.

Reply: As suggested, we provided the Figure and Table (Editor suggested) about the topics.

  1. Is recommendable focus only in metabolic syndrome, and discuss meticulously about the molecular mechanisms that aporphine alkaloids regulate.

Reply: Thank you for your kind advice. Metabolic syndrome as a clustering disorder includes abnormal insulin and glucose metabolism, insulin resistance, hyperglycemia, hyperlipidemia, hypertension, hyperuricemia, and oxidative stress and pro-inflammatory state. Metabolic syndrome is defined as a set of risk factors for various diseases. Therefore, in this review, we focus on the impact of aporphine alkaloids on metabolic syndrome and its associated disorders. We carefully re-read each original study, but many studies only report the phenomenon, thus, it is difficult to discuss meticulously about the molecular mechanisms that aporphine alkaloids regulate in this review. In each paragraph, we try to discuss after re-reading literatures. In addition, insulin resistance is the major underlying mechanism responsible for metabolic syndrome. We also stated this in the conclusion. Almost aporphine alkaloids in this review can prevent insulin resistance or/and regulate glucose homeostasis, of which boldine and nuciferine have potential pharmacological effects on metabolic syndrome components. More information was provided in the revised manuscript.

  1. There is a structural association about the protein-targets that are regulated? An analysis is desirable.

Reply: Thank you for the advice. After re-reading literatures, there is not a structural association about the protein-targets that are regulated. But there is the molecular docking analysis for some aporphine alkaloids, which were provided in the text and Table of the revised manuscript.

  1. It is desirable including a scheme with the structures of analyzed aporphine alkaloids.

Reply: As suggested, the structure of natural aporphine alkaloids with potential to impact metabolic syndrome was shown in the Figure. Insulin resistance is the major underlying mechanism responsible for metabolic syndrome. Of note, almost aporphine alkaloids in this review can prevent insulin resistance or/and regulate glucose homeostasis, of which boldine and nuciferine  have  potential pharmacological effects on metabolic syndrome components (such as endothelial dysfunction, hypertension and cardiovascular disease, hyperlipidemia and obesity, non-alcoholic fatty liver disease, hyperuricemia and kidney damage, erectile dysfunction, central nervous system-related disorder, intestinal microbiota dysbiosis). Thus, the aporphine core structure with methoxyl or/and hydroxyl substitutions at its C1, C2, C9 or/and C10 positions may be important for these activities (Figure). However, the structure-activity profiling of natural aporphine alkaloids against metabolic syndrome and its components should be addressed in the future. The detail information was provided in the revised manuscript.

Reviewer 2 Report

The authors reviewed not only the effects of aporphine alkaloids on metabolic syndrome and related diseases but also their potential mechanisms citing recent papers. This review would be informative for many readers.

The reviewer raised minor comment:

Please reconsider the sectioning of manuscript flow. Dyslipidemia is closely related to vascular and hepatic functions. While the author discussed hyperlipidemia in section 4, the topics of dyslipidemia straddle several sections. This overlap would lead to confusion in understanding of readers.

Author Response

In response to Comments and Suggestions for Authors
-Reviewer 2

The authors reviewed not only the effects of aporphine alkaloids on metabolic syndrome and related diseases but also their potential mechanisms citing recent papers. This review would be informative for many readers.

The reviewer raised minor comment:

  1. Please reconsider the sectioning of manuscript flow. Dyslipidemia is closely related to vascular and hepatic functions. While the author discussed hyperlipidemia in section 4, the topics of dyslipidemia straddle several sections. This overlap would lead to confusion in understanding of readers.

Reply: Thank you for the suggestions. Lipid profile change is observed in patients with metabolic syndrome and other related dysfunction, for the sake of completeness, some of aporphine alkaloids on dyslipidemia were reserved briefly in the several sections. As suggested, we partly adjusted and revised the Dyslipidemia in several sections to reduce the overlap.

The revision was marked in red in the revised manuscript. We hope that this revision successfully address the concerns, and this manuscript is accepted for publication.

Thank you and best regards.

Looking forward to hearing from you soon.

Yours sincerely,

Mr. Fei-Xuan Wang

Nanjing Institute of Product Quality Inspection

Nanjing 210019

China

Email: feixuanwang@qq.com, cyruswangc@gmail.com

Tel.: +86-25-86673778

Round 2

Reviewer 1 Report

I recommend you review some typos through the manuscript.

Congratulations!

Best regards,